# Fludarabine–Cyclophosphamide-Based Conditioning with Antithymocyte Globulin Serotherapy Is Associated with Durable Engraftment and Manageable Infections in Children with Severe Aplastic Anemia

**DOI:** 10.3390/jcm10194416

**Published:** 2021-09-26

**Authors:** Małgorzata Salamonowicz-Bodzioch, Monika Rosa, Jowita Frączkiewicz, Ewa Gorczyńska, Katarzyna Gul, Małgorzata Janeczko-Czarnecka, Tomasz Jarmoliński, Krzysztof Kałwak, Monika Mielcarek-Siedziuk, Igor Olejnik, Joanna Owoc-Lempach, Anna Panasiuk, Kornelia Gajek, Blanka Rybka, Renata Ryczan-Krawczyk, Marek Ussowicz

**Affiliations:** 1Department of Pediatric Oncology, Haematology and Bone Marrow Transplantation, Wroclaw Medical University, Borowska 213, 50-556 Wroclaw, Poland; rosaa.monika@gmail.com (M.R.); jowitafr@gmail.com (J.F.); ewa.gorczynska@gmail.com (E.G.); katarzynagul@o2.pl (K.G.); ml.janeczko@gmail.com (M.J.-C.); tjarmo@wp.pl (T.J.); krzysztof.kalwak@gmail.com (K.K.); m.mielcarek@gmail.com (M.M.-S.); olejnik@olejnik.x.pl (I.O.); owocowa@interia.pl (J.O.-L.); kornelia_gajek@hotmail.com (K.G.); blankarybka@interia.pl (B.R.); renataryczan@interia.pl (R.R.-K.); ussowicz@gmail.com (M.U.); 2Department of Oncology, Hematolgy and Transplantology, University Hospital USK in Wroclaw, 50-556 Wroclaw, Poland; annapanasiuk@yahoo.pl

**Keywords:** aplastic anemia, fludarabine–cyclophosphamide-based conditioning, ATG serotherapy, hematopoietic cell transplantation, children, viral infections, viral replication

## Abstract

Severe aplastic anemia (SAA) is a bone marrow failure syndrome that can be treated with hematopoietic cell transplantation (HCT) or immunosuppressive (IS) therapy. A retrospective cohort of 56 children with SAA undergoing transplantation with fludarabine–cyclophosphamide–ATG-based conditioning (FluCyATG) was analyzed. The endpoints were overall survival (OS), event-free survival (EFS), cumulative incidence (CI) of graft versus host disease (GVHD) and CI of viral replication. Engraftment was achieved in 53/56 patients, and four patients died (two due to fungal infection, and two of neuroinfection). The median time to neutrophil engraftment was 14 days and to platelet engraftment was 16 days, and median donor chimerism was above 98%. The overall incidence of acute GVHD was 41.5%, and that of grade III-IV acute GVHD was 14.3%. Chronic GVHD was diagnosed in 14.2% of children. The probability of 2-year GVHD-free survival was 76.1%. In the univariate analysis, a higher dose of cyclophosphamide and previous IS therapy were significant risk factors for worse overall survival. Episodes of viral replication occurred in 33/56 (58.9%) patients, but did not influence OS. The main advantages of FluCyATG include early engraftment with a very high level of donor chimerism, high overall survival and a low risk of viral replication after HCT.

## 1. Introduction

Severe aplastic anemia (SAA) is a rare but life-threatening hematological disorder with an extremely high risk of fatal infectious complications. The hallmark of SAA is pancytopenia caused by bone marrow (BM) hypoplasia or aplasia as a consequence of direct damage by chemical or physical factors or constitutional or acquired genetic defects, e.g., Fanconi anemia, or telomere biology disorders [1,2,3]. In the majority of SAA patients, the cause cannot be directly identified, but immune-mediated destruction of BM hematopoiesis is the most likely culprit. Autoimmunity can be triggered by alterations in antigens modified by drugs, chemical agents or viral infections and, consequently, can lead to the activation of the immunological cascade and damage to BM cells via activated T lymphocytes [2,4,5,6]. Allogeneic hematopoietic cell transplantation (HCT) from HLA-identical matched sibling donors (MSDs) is the method of choice for the treatment of SAA in children. Patients without sibling donors undergo immunosuppressive (IS) therapy or HCT from matched unrelated donors (MUDs) [6,7,8,9]. In the last 20 years, different protocols have been used in clinical practice in our center, and, at that time, in Polish children with SAA, MSD and MUD HCT were associated with a 5-year probability overall survival of 91 and 64%, respectively [10]. Treosulfan-based conditioning before MUD HCT was associated with 35% treatment-related mortality (Figure 1), and a decision was reached to introduce a combination of fludarabine, cyclophosphamide and antithymocyte globulin (FluCyATG). It must be mentioned here that patients who underwent transplantation after treosulfan had a median time of 19 months from diagnosis and a longer prior history of blood transfusions. The aim of this study was to evaluate the outcome of a conditioning protocol, FluCyATG, in children undergoing transplantation for SAA.

## 2. Materials and Methods

The group consisted of 56 children (aged 0.8–17.9 years) with a diagnosis of SAA who underwent HCT with transplants from MSDs or MUDs. The patient characteristics are shown in Table 1.

Patients with identified constitutional syndromes and those who underwent retransplantation were not included in the analysis. The original conditioning protocol, given in 2008–2011 to 12 patients, consisted of fludarabine at a dose of 40 mg/m^2^ on days −8 to −5, cyclophosphamide at a dose of 50 mg/kg BW/day on days −5 to −1 and Thymoglobulin (Sanofi) at a dose of 2.5 mg/kg BW daily on days −4 to −1. The modified conditioning protocol, administered after 2011 in 44 patients, consisted of fludarabine at a dose of 30 mg/m^2^ on days −6 to −3 and cyclophosphamide at a dose of 750 mg/m^2^/day on days −6 to −3. From day −6 to day −3, antithymocyte globulin Grafalon (Neovii) at 15 mg/kg BW daily or Thymoglobulin (Sanofi) at 2.5 mg/kg BW daily was given. Peripheral blood stem cells (PBSCs) were used as a stem cell source in 31 out of 38 (82%) patients receiving a transplant from MUDs and in 6 of 17 (36%) patients receiving a transplant from MSDs; the remaining patients underwent BM transplantation. Graft versus host disease (GVHD) prophylaxis was based on ciclosporin A started on pretransplantation day –1 and methotrexate 15 mg/m^2^ given on posttransplantation days +1, +3 and +6. Antimicrobial guidelines are summarized in Appendix B. The patients or legal guardians gave their written informed consent for the treatment and analysis of clinical data. Ethical approval was waived by the local Ethics Committee of Wroclaw Medical University in view of the retrospective nature of the study and because all procedures were performed as a part of routine care.

### Statistical Analysis

The endpoints were overall survival (OS), defined as the time from HCT to death or the last report from patients with no events, and event-free survival (EFS), defined as the time from HCT to graft rejection, second malignancy or death. Because no patients experienced events except deaths, the OS and EFS results were the same, and only OS data are presented here. GVHD-free survival (GFS) was defined as the absence of grade III-IV acute GVHD, chronic GVHD that required systemic treatment and death, similar to the composite endpoints proposed by Holtan [11].

Survival curves were estimated using the Kaplan–Meier method and compared between the cohorts by the log-rank test. Cox modeling was adopted to estimate hazard ratios for OS and EFS, considering factors with *p* < 0.2. Statistical analysis and data formatting for presentation were performed with the GraphPad Prism software (GraphPad Software, La Jolla, CA, USA) and STATISTICA 13.3 (TIBCO Software Inc. 2017, STATISTICA, version 13, Dell, OK, USA).

## 3. Results

All survival results in the study cohort are presented in Table 2. Three patients died before engraftment on days +1, +5 and +17. Two patients died of uncontrolled invasive fungal infection (one of mucormycosis and one of aspergillosis), and one died as a consequence of acute neurotoxicity of undetermined background.

The remaining patients achieved trilineage bone marrow recovery, and no secondary graft failure was found. The median time to absolute neutrophil count over 500/µL was 14 days (range 10–22 days), and the median time to platelet count over 20000/µL was 16 days (range 5–212 days) (Figure 2A).

Median donor chimerism at 1, 3, 12 and 24 months after HCT was 98%, 99%, 100% and 100%, respectively (Figure 2B). Among patients whose transplants engrafted, the only observed death was due to neuroinfection.

The overall incidence of acute GVHD was 41.5%, and grade III-IV aGVHD was diagnosed in 14.3% of patients. Chronic GVHD was diagnosed in 14.2% of children within 2 years of HCT, and the grade was moderate to severe in 10.4% (Figure 3B). The probability of 2-year GVHD-free survival was 76.1% (Figure 3C). In the univariate analysis, the antecedent IS protocol was the only factor associated with a significantly lower probability of overall survival (83.3 vs. 100%, *p* = 0.0017, Figure 3D). The reduction in the intensity of the FluCyATG protocol was associated with improved survival (97.7% vs. 85%, *p* = 0.0063, Figure 3E). Gender-related differences between donor and recipient affected survival (Figure 3E), with a significantly lower probability of survival with female donors (80% vs. 100%, *p* = 0.005, Figure 3F). The overall and graft-free survival curves of all studied factors associated with the HCT procedure are presented in Appendix A.

In addition, the probability of OS was analyzed in the two subgroups: 18 patients received transplants from MSDs after FluCyATG and a historic group of 13 patients received transplants from MSDs after cyclophosphamide 200 mg/kg BW with or without ATG; the probability of 5-year OS was 94.1 and 100%, respectively, but the difference was not statistically significant (Figure 4A). The percentage of donor chimerism between these two groups was not significantly different 1, 3, 12 and 24 months after HCT (Figure 4B).

Posttransplantation viral replications were observed in 33/56 patients (58.9%). ADV viremia was found in 12.5% of patients, BKV in 28.6% of patients, CMV in 28.6% of children and EBV replication in 21.4% of children. In all patients, viral replication was asymptomatic, and preemptive treatment prevented the development of clinical manifestations. The presence of posttransplantation viral infections did not influence OS or GFS (Appendix A).

In the Cox multivariate analysis, none of the analyzed factors were significantly associated with either OS or GFS.

## 4. Discussion

The results of SAA therapy in children are superior to those in the adult population, but the role of IS therapy in recent years has diminished in favor of upfront transplantation from matched unrelated donors due to the high IS failure rate, with an EFS of 33% [12]. Survival in pediatric SAA has improved dramatically for MUD HCT, due to improvements in donor typing, less toxic conditioning regimens with low-dose TBI or TBI free and the use of leukodepleted blood products [13,14,15]. The OS is higher in children who receive a transplant from an MSD than in those who receive a transplant from an MUD (96% vs. 91%, *p* = ns), but the long-term outcome and freedom from cGVHD have not been properly analyzed [12,16].

In our study, the HCT results in patients with SAA after FluCyATG conditioning were unquestionably good in terms of neutrophil engraftment, predominant donor chimerism and overall survival. The FluCyATG conditioning protocol was associated with an OS comparable to that of the gold-standard cyclophosphamide–ATG (CyATG) in MSD HCTs. The reduction in cyclophosphamide dose in our cohort was balanced by the effect of fludarabine, and no detrimental effect on the probability of OS or donor chimerism level was observed. Among different factors affecting OS, an impact of female donors (independent from recipient sex) was associated with inferior survival. Among non-HLA donor characteristics, sex mismatching (male recipient–female donor) is a proven risk factor for inferior survival associated with cGVHD incidence, but due to the low number of events in our study, this result must be approached with caution [17,18,19].

The introduction of fludarabine into the conditioning regimens of adults with SAA has been seen in the last 20 years [20]. FluCyATG with a cyclophosphamide dose of 200 mg/kg BW is more effective in terms of overall survival and engraftment than CyATG [21]. In the EBMT study, after using fludarabine at a dose of 120 mg/m^2^, cyclophosphamide at a dose of 1200 mg/m^2^ (40 mg/kg) and Thymoglobulin at a dose of 15 mg/kg BW, the actuarial 2-year OS was 72%, but graft rejections were observed, and posttransplantation mortality was associated with infections and GVHD [22]. A study by Resnick et al. reported an OS of 84% after fludarabine 180 mg/m^2^, cyclophosphamide 120 mg/kg and ATG (total dose 40 mg/kg BW) [23]. A Korean study showed improved OS and lower toxicity in patients with SAA who underwent transplantation and received cyclophosphamide at a dose of 120 mg/kg BW than in those who received a dose of 200 mg/kg BW [24].

The influence of the cyclophosphamide dose reduction in our study on the incidence of long-term sequelae was not proven due to the insufficient period of observation, but the reduction in alkylator dose can be expected to be beneficial in terms of short- and long-term toxicity. Impaired spermatogenesis is unlikely when the cyclophosphamide dose is less than 4000 mg/m^2^, and impaired oocytogeneis is unlikely when the cyclophosphamide is below 6000 mg/m^2^ [25,26]. However, the effect of fludarabine on fertility outcomes needs to be assessed further [27].

Notably, FluCyATG was highly effective in a subgroup of patients showing PNH-positive clonal disease. This experience encourages transplantation in the pediatric subtype of PNH (+) SAA with less intensive protocols than the treosulfan-based regimens used in the adult population [28]. No patient in our group showed posttransplantation clonal disease. HCT has an advantage over IS therapy through its reduced risk of myelodysplasia and leukemic transformation, although this phenomenon was mostly reported in the adult SAA population [29].

The question of the best GVHD prophylaxis is yet unanswered, but ATG is widely used in SAA. ATG has a narrow therapeutic window and therapeutic drug monitoring of ATG levels helps to optimize dosing to ensure timely T-cell immune reconstitution [30,31]. Exposure to ATG affects survival after HCT in adults, highlighting the importance of optimum ATG dosing. According to these studies, overexposure of ATG delays T-cell reconstitution and is associated with increased relapse rates and viral reactivations, whereas underexposure is associated with the incidence of GvHD and higher mortality [32]. Individualized dosing of ATG, based on lymphocyte counts rather than bodyweight, in adults has been recommended by Admiraal at al., but in children, higher exposure can be observed in patients with a higher bodyweight and/or a lower lymphocyte count pre-Thymoglobulin infusion [30,32]. The results of a mixed pediatric and adult study supported FluCyATG and CyATG as optimal regimens for MSD BMT and the use of rabbit-derived ATG in MUD settings due to a lower risk of acute GVHD [33]. Serotherapy with rabbit ATG, equine ATG or alemtuzumab in SAA transplantation settings was shown to be associated with a survival advantage, but studies also support different approaches [34]. In the randomized study by Champlin, the 5-year overall probabilities of survival after alloHCT from MSD were 74% after cyclophosphamide alone and 80% after cyclophosphamide and equine ATG, but the difference did not reach statistical significance [35]. Another serotherapy option is the administration of alemtuzumab. In a study by Marsh et al., the conditioning regimen consisted of fludarabine at a dose of 30 mg/m^2^ for 4 days, cyclophosphamide at a dose of 300 mg/m^2^ for 4 days and alemtuzumab at a median total dose of 60 mg. OS was 95% and 83% in the MSD and MUD subgroups, respectively [36]. In this study, graft failure occurred in 12% of patients, and no evaluated patient achieved full donor chimerism in T lymphocytes, which is suboptimal because idiopathic SAA is caused by oligoclonal T lymphocytes eliminating hematopoietic cells [37,38]. The administration of the same protocol in a study by Shah was associated with 100% survival, and all patients had full donor (>95%) myeloid chimerism from the 3rd month post-HCT until the last follow-up, but one third of children undergoing transplantation showed less than 50% donor cells among their T lymphocytes [39]. In one of the largest studies to date, Dufour et al. reported a 96% probability of survival in MUD patients, 91% in MSD controls and 74% survival after failed IS, which is in line with our results [12]. The conditioning regimen used in the study consisted of fludarabine 150 mg/m^2^, cyclophosphamide 120 mg/kg BW and alemtuzumab 0.9–1 mg/kg BW. Seventy-two percent of patients received BM transplants, which is different from our group. The 1-year CI of grade II–IV aGVHD was 10 ± 6%. There was only one case of grade III/IV aGVHD (frequency of 3.5%; in one patient receiving MUD HCT) requiring systemic immunosuppression with steroids. The 1-year CI of cGVHD was 19 ± 8% in the Dufour study, and all cases showed only limited grade GVHD with skin involvement. The median whole-blood donor chimerism at the last follow-up was 100% (range 88–100%)^11^. Viral infections constitute a significant cause of morbidity and mortality after HCT [40]. Dufour et al. highlighted in his study that viral reactivation was common, accounting for 49%, but no fatal outcome was reported [12]. In contrast, Im et al. observed a lower incidence of viral infections (23%), but some of them led to death, and viral reactivation was found to be an independent risk factor for lower OS in this study [41]. Similar results were reported in a Pakistani study, where any viremia was reported in 30% of children after FluCyATG, and mortality was observed in 2.7% of patients [42]. In contrast, Kang et al. recorded that 89% of patients developed viral reactivation, but only two of them died due to viral infection [43]. In our study, the incidence of the replication of any virus was 59%, but there was no impact on survival. This effect can be explained by regular viral surveillance and timely preemptive treatment.

The most common viral reactivations in our cohort were asymptomatic CMV and BKV, both detected in 28.9% of patients. These results are in line with Chaudhry et al., who detected an incidence of 27.6% for CMV infections [42]. In contrast, Im et al. recorded a lower rate of CMV reactivation—20.9%—but CMV reactivation was associated with 3% of fatalities [41]. Kang et al. observed a CMV CI of 69.1% in patients with SAA after HCT but did not describe deaths connected directly to CMV [43]. Dufour et al. observed CMV viremia in 17.2% of patients, and no fatal cases were described [12]. The incidence of BKV seems to be low among children with SAA after HCT [44]. In children with malignancies, the incidence of BKV was shown to be 25–62%, and up to 27% patients developed BKV-hemorrhagic cystitis [45]. Malignant indications were reported in multiple studies to be associated with symptomatic BKV infection, but this can be explained by a more intensive conditioning protocol with uroepithelial damage and intensive immune suppression [45,46,47]. In our cohort, despite BKV replication in 29% of HCT recipients, the infections were asymptomatic and not associated with death. In contrast to our results, Chaudhry observed BKV replication in only 2.7% of patients [42]. Similarly, Giraud et al. observed that hemorrhagic cystitis and BK viruria were less common in patients receiving RIC than in those receiving full conditioning [44].

The risk of ADV infections is increased among pediatric patients, and higher infection rates have been reported over the last few decades [48]. We reported an ADV incidence of 12.5%, which was lower than the 25–50% reported in other studies [49,50]. Moreover, patients were asymptomatic, and ADV viremia and/or viruria did not influence survival. Interestingly, in a study by Dufour et al., in children with SAA who underwent transplantation and received the FCC protocol, the ADV incidence rate was only 3.4% [12]. We found EBV reactivation in 21% of cases but without overt PTLD. A Korean study reported only one case of EBV reactivation followed by PTLD among 43 patients [41]. In a study by Kang, EBV CI was 17.8%, and in 7% of cases, EBV ended with death due to progressive PTLD [43]. In contrast, Dufour et al. reported EBV replication in 10.3% of patients [12]. These differences might result from multiple factors, such as different ages, conditioning regimens, local epidemiologies, seasonal factors and viral testing methods.

Another factor that can have a decisive role in the referral for HCT is the incidence of life-threatening or severely disabling acute or chronic GVHD. In our study, we observed a 76.1% probability of GFS. This result can be influenced by the increasing trend in PBSC transplantations from MUDs due to the convenience of graft processing. However, according to the EBMT analysis, the use of peripheral blood grafts in pediatric HCTs remains the strongest negative predictor of survival [34]. PBSC is not considered the best stem cell source for the first HCT in SAA because of the lower survival and higher risk of aGVHD and cGVHD [34,51,52]. However, more recent studies report better outcomes and similar results among MUDs with PBSC or BM [12]. PBSC has been reported in cases of a second donation by the same donor where previously BM was used to improve engraftment and achieve faster hematopoietic recovery [53]. Horan et al. reported the incidence of graft failure to be 43% in 166 SAA patients who underwent second HCT from MSD using BM or PBSC in 84 and 16% of patients, respectively [54]. Similarly in a pediatric study by Cesaro concerning second allogenic transplantation, use of PBSC was not associated with inferior OS [55].

The observed GFS results can be seen as the main concern challenging the HCT, because, in contrast to adult patients, children’s life expectancy is 50–70 years, and the risk of cGVHD-associated organ complications and metabolic consequences of life-long steroid immunosuppression must not be neglected. The place for GFS improvement can be found in the preferred source, BM, proven to be associated with a lower risk of GVHD in children with acute lymphoblastic leukemia [56]. It must be emphasized that our study did not reveal worse results in patients receiving PBSC transplants than in those receiving BM transplants, and minimal tissue damage by FluCyATG can be suggested as another factor decreasing the induction phase of GVHD [57].

Finally, the conundrum that needs to be resolved in pediatric SAA is the question of when the patient should be referred for HCT and whether upfront HCT is better than IS. Our results show that a median time to transplantation of 4.5 months leaves a window of opportunity for a nontransplant strategy before the availability of the alternative donor. The chance of improving IS results can be seen in using new drugs, such as thrombopoietin receptor antagonists, which have been shown to stimulate the proliferation of autologous hematopoietic stem cells, resulting in the licensing of eltrombopag in SAA in adults and children over 12 years of age, but studies in children have not confirmed the efficacy of SAA upfront therapy [58,59].

## 5. Conclusions

The benefits of FluCyATG are associated with high overall survival probability, early engraftment with a very high level of donor chimerism and minimal impact of posttransplantation opportunistic infections. Viral monitoring and timely preemptive treatment can reduce the impact of posttransplantation infections to a clinically irrelevant factor. The high curability of SAA raises the issue of long-term sequelae reduction, which can be resolved by less toxic conditioning protocols and prudent referral for alloHCT. The waiting time for alternative-donor HCT in SAA can still be used as a window for pharmacotherapy involving IS and thrombopoietic drugs.

## Figures and Tables

**Figure 1 jcm-10-04416-f001:**
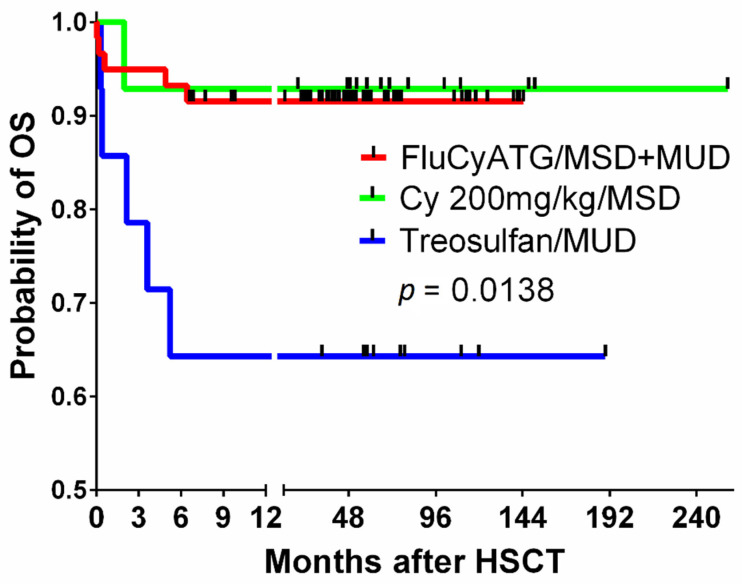
The overall survival of children with SAA undergoing transplantation after the fludarabine–cyclophosphamide–ATG protocol compared to that of a historic group of MSD recipients conditioned with cyclophosphamide 200 mg/kg BW and ATG and MUD recipients after treosulfan-based conditioning. Legend. SAA, severe aplastic anemia; ATG, antithymocyte globulin, MSD, matched sibling donor; BW, body weight; MUD, matched unrelated donor.

**Figure 2 jcm-10-04416-f002:**
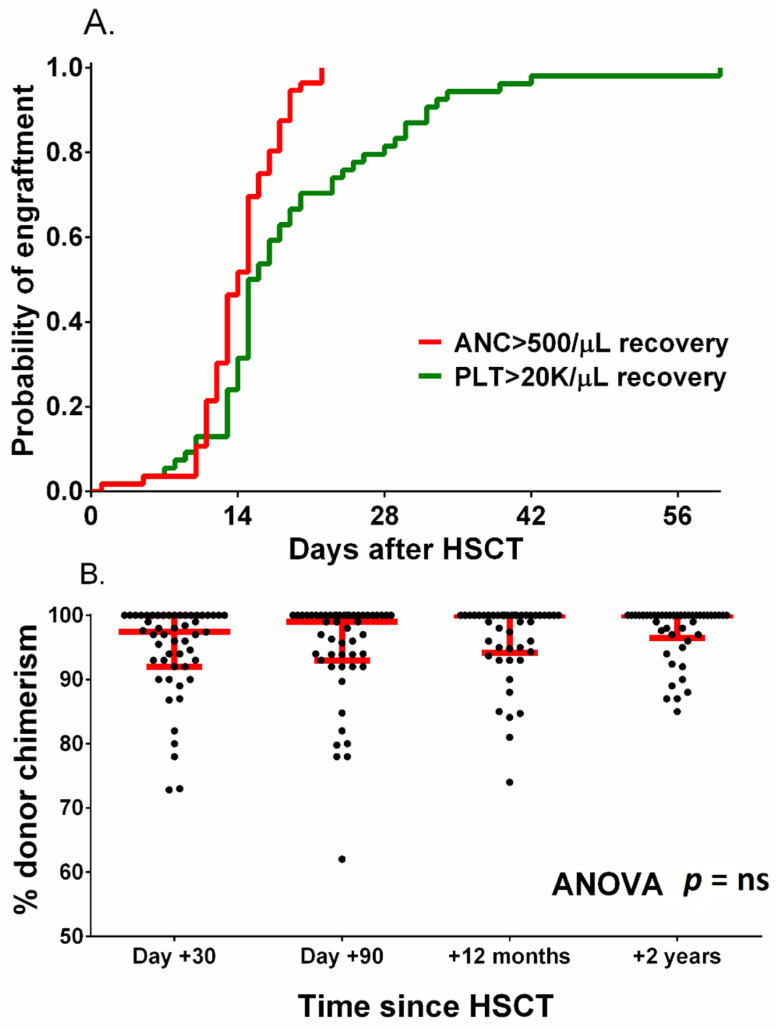
Probability of neutrophil and platelet engraftment (**A**) and donor chimerism after HCT (**B**). Red lines in figure (**B**) represent the median with interquartile range.

**Figure 3 jcm-10-04416-f003:**
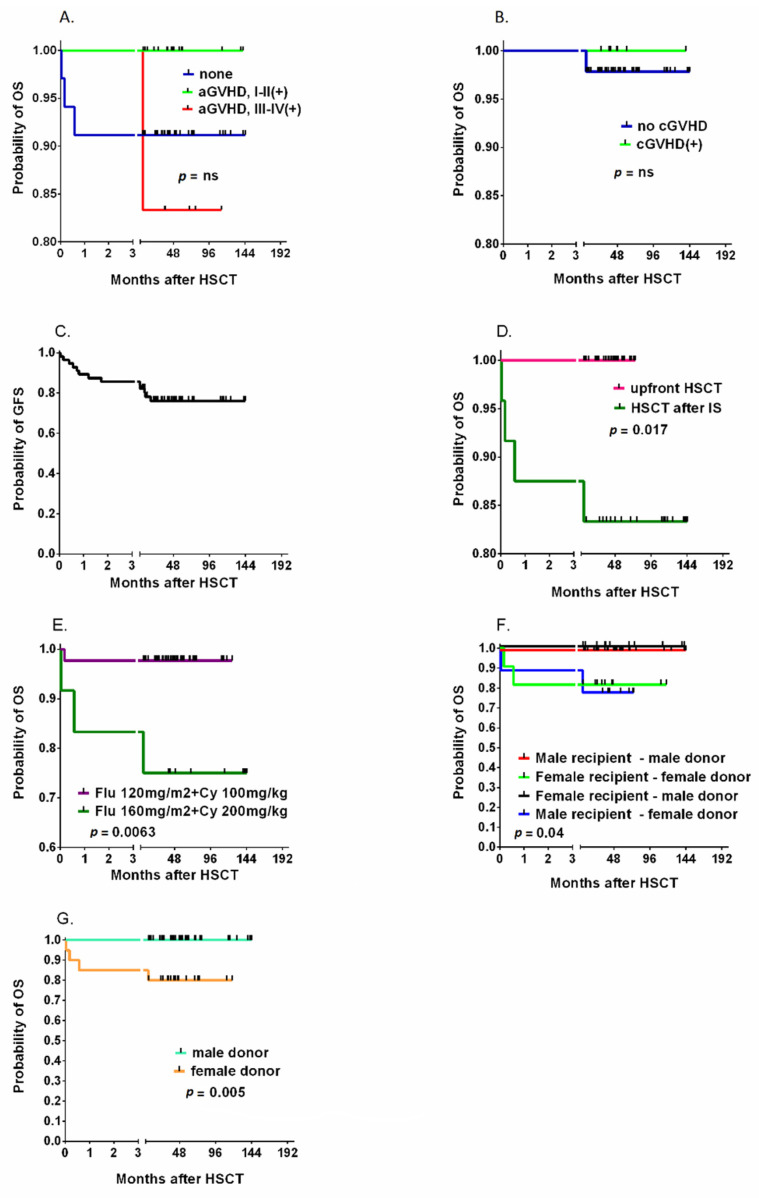
The cumulative incidence of acute GVHD (**A**) and chronic GVHD (**B**) and the probability of graft-free survival (**C**) after HSCT. The impact of pretransplantation immunosuppressive therapy (IS) (**D**) and the intensity of the chemotherapy protocol (**E**) on the probability of OS. The role of recipient and donor genders (**F**) and of donor genders (**G**). Legend. OS, overall survival; GFS, graft versus host disease-free survival; HSCT, Allogeneic hematopoietic cell transplantation; Flu, fludarabine; Cy, cyclophosphamide; GVHD, graft versus host disease; aGVHD, acute graft versus host disease; cGVHD, chronic graft versus host disease.

**Figure 4 jcm-10-04416-f004:**
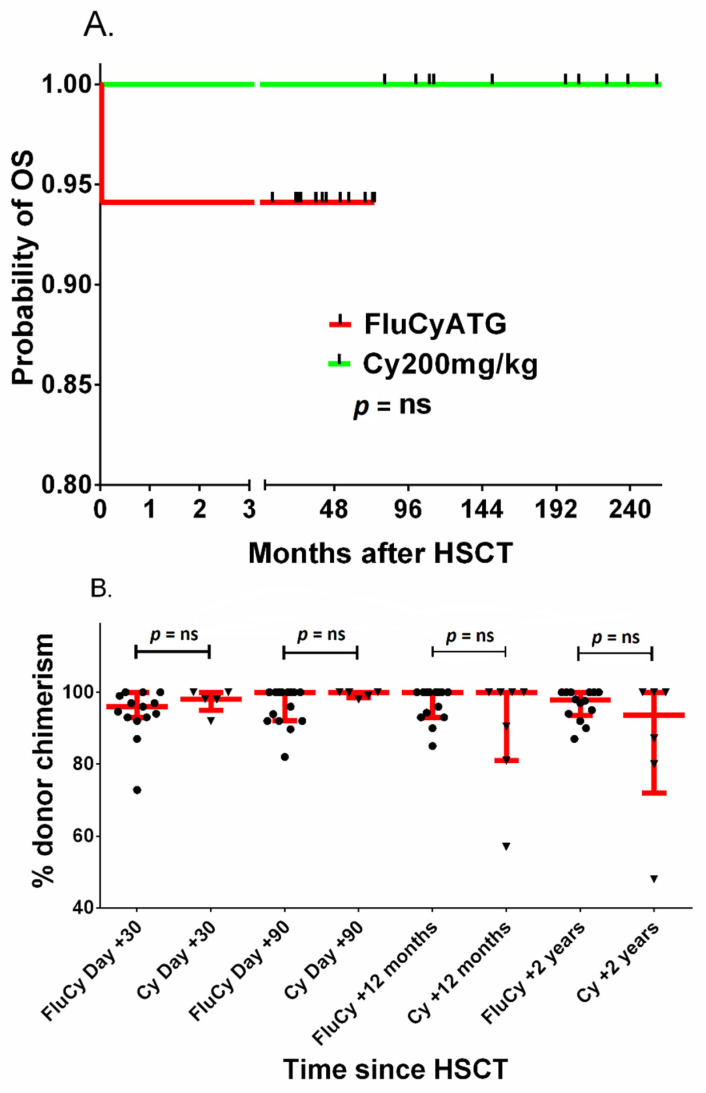
Probability of OS (**A**) and donor chimerism (**B**) after HCT with fludarabine–cyclophosphamide–ATG (FluCyATG) or cyclophosphamide 200 mg/kg BW. Red lines in figure (**B**) represent the median with interquartile range. Legend. OS, overall survival; GFS, graft versus host disease-free survival; HSCT/HCT, Allogeneic hematopoietic cell transplantation; Flu, fludarabine; Cy, cyclophosphamide; GVHD, graft versus host disease; aGVHD, acute graft versus host disease; cGVHD, chronic graft versus host disease ATG, antithymocyte globulin.

**Table 1 jcm-10-04416-t001:** Patient and transplantation characteristics.

Category		Value
Number of Patients		56
Sex	male	33
	female	23
Age at HCT in years	median	9.4
	range	0.8–17.8
Time from diagnosis to HCT in months	median	4.57
	range	1.23–66.6
Previous IS protocol	yes	14
	no	32
PNH clone	detectable	8
	undetectable	48
Donor	matched sibling donor	18
	matched unrelated donor	38
Degree of HLA match	matched sibling donor, 6/6	15
	matched sibling donor, 10/10	2
	matched unrelated donor, 10/10	33
	matched sibling/unrelated donor, 9/10	6
Stem cell source	bone marrow	17
	peripheral blood stem cells	39
Bone marrow CD34 + cells/kg in millions	median (range)	3.54 (0.34–12.14)
Peripheral blood CD34 + cells/kg in millions	median (range)	9.31 (2.42–29.78)
Antithymocyte globulin	Grafalon	15
	Thymoglobulin	41
Posttransplantation methotrexate	yes	54
	no	2
Posttransplantation follow-up in months	median	44
	range	0–144
Time to neutrophil count > 500/µL in days	median	14
	range	10–22
Time to platelet count > 20,000/µL in days	median	16
	range	5–212
Acute graft versus host disease	any grade	22
	grade III-IV	6
Chronic graft versus host disease	any grade	7
	moderate–severe grade	5

**Table 2 jcm-10-04416-t002:** Impact of factors affecting survival after HCT.

Category		Number of Patients	5-Year OS	Log-Rank *p*	2-Year GFS	Log-Rank *p*
Sex	male	33	93.9	ns	84.5	ns
	female	23	91.3		63.4	
Previous IS protocol	yes	14	83.3	*p* = 0.017	66.4	ns
	no	32	100		83.5	
Time from diagnosis to HCT	<3 months	14	100	ns	91.7	ns
	>3 months	41	90.2		70.4	
PNH clone	detectable	8	100	ns	100	ns
	undetectable	48	91.7		72	
Donor	matched sibling donor	18	94.1	ns	87.8	ns
	matched unrelated donor	38	92.3		71	
Recipient–donor gender	Male donor–male recipient	24	100	*p* = 0.04	86.8	ns
	Female donor–female recipient	11	81.8		53	
	Male donor–female recipient	9	77.8		77.8	
	Female donor–male recipient	12	100		72.9	
Donor gender	Male donor	36	100	*p* = 0.005	82.5	ns
	Female donor	20	80		64.6	
CMV IgG status	Donor positive–recipient positive	28	90.3	ns	73.1	ns
	Donor positive–recipient negative	4	100		100	
	Donor negative–recipient positive	15	93.8		75	
	Donor negative–recipient negative	4	100		75	
Stem cell source	bone marrow	17	88.2	ns	69.7	ns
	peripheral blood stem cells	39	94.9		79	
Conditioning	Flu 160 mg/m^2^ + Cy 200 mg/kg BW	12	85	*p* = 0.0063	58.3	ns
	Flu 120 mg/m^2^ + Cy 100 mg/kg BW	44	97.7		81	
Antithymocyte globulin	Grafalon	15	100	ns	100	ns
	Thymoglobulin	41	90.2		70.7	
Acute graft versus host disease	no	34	91.2	ns	n/a	
	grade I-II	16	100			
	grade III-IV	6	83.3			
Chronic graft versus host disease	no	49	97.8	ns	n/a	
	any grade	7	100			
Posttransplantation ADV replication	yes	7	100	ns	76.9	ns
	no	49	91.8		71.4	
Posttransplantation BKV replication	yes	16	100	ns	87.5	ns
	no	40	90		71.9	
Posttransplantation CMV replication	yes	16	100	ns	93.8	ns
	no	40	90		69.1	
Posttransplantation EBV replication	yes	12	91.7	ns	74.1	ns
	no	44	93.2		76.7	

**Legend.** OS, overall survival; GFS, graft versus host disease-free survival; HCT, Allogeneic hematopoietic cell transplantation; PNH, paroxysmal nocturnal hemoglobinuria; Flu, fludarabine; Cy, cyclophosphamide.

## Data Availability

The data presented in this study are available on request from the corresponding author.

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
