# Peer review of "Fludarabine–Cyclophosphamide-Based Conditioning with Antithymocyte Globulin Serotherapy Is Associated with Durable Engraftment and Manageable Infections in Children with Severe Aplastic Anemia"

_jcm, 2021, doi:10.3390/jcm10194416_

Round 1
Reviewer 1 Report
Very interesting study and results. Did you note any potential associations between patients who developed chronic GVHD versus the patients who did not develop chronic GVHD outside of increase in peripheral blood stem cell transplantations from match unrelated donors?
Author Response
Response to Reviewer 1 Comments
General response: Thank you for your review. The manuscript was modified according to the reviewers requests. To address the reviewer’s remarks, we added information to tables 1 and 2, presented additional figures (3 F, G) and revised supplemental figures. In addition we changed from black and white figures to color figures, to increase their legibility.
Very interesting study and results.
- Did you note any potential associations between patients who developed chronic GVHD versus the patients who did not develop chronic GVHD outside of increase in peripheral blood stem cell transplantations from match unrelated donors?
No significant factors were associated with a/cGVHD incidence in the study group. Please, note that PBSC (or any other factor) were not associated with different risk of graft-versus-host free survival. This observation might support the hypothesis, that this transplant protocol reduces risk of GVHD.

Reviewer 2 Report
This is a well written report on HCT which is easy to read in a cohort of pediatric patients with SAA.The outcome is expected and in line with what is achieved in most HCT centers today.It is more or less standard of care.
Some points and suggestions to improve the manuscript;
1.Survival has improved following HCT during the last decades,which has been published from singel and multicenter reports.This should be discussed.
2.What was the degree of HLA matches in the recipients of MUD?Ie how many were 10/10,9/10 etc.
3.How come PBSC was used as graft sorce in SAA patients,who do not benefit from chronic GVHD in contrast to patients with malignant disorders.BM should be preferred in patients with SAA and other non-malignant diseases.Refer to studies showing a higher risk of chronic GVHD using PBSC vs BM and discuss pros and cons in pediatric HCT for SAA.
4.Please give more information in the table on characteristics like CMV status of recipients and donors,nucleated cell dose,CD 34+ celldose,sex match of donor and recipients etc.Such data is of interest for the reader to evaluare the results.
5.Regarding ATG,Thymoglobuline treatment,the patients were given different dosing and timing up to day -1 and day -3.Please discuss the importance of dosing and timing of Thymoglobuline with adequate references.This may affect risks of acute GVHD and infections respectively.
Minor, change HSCT to HCT,because this is not a true stemcell transplant.
Supplementary data may be deleted,because they were ns.Furthermore they were not possible to download.
Author Response
Response to Reviewer 2 Comments
Reviewer 2
General response: Thank you for your review. The manuscript was modified according to the reviewers requests. To address the reviewer’s remarks, we added information to tables 1 and 2, presented additional figures (3 F, G) and revised supplemental figures. In addition we changed from black and white figures to color figures, to increase their legibility.
This is a well written report on HCT which is easy to read in a cohort of pediatric patients with SAA. The outcome is expected and in line with what is achieved in most HCT centers today. It is more or less standard of care.
Some points and suggestions to improve the manuscript;
- Survival has improved following HCT during the last decades, which has been published from singel and multicenter reports. This should be discussed.
We are discussing this issue in the article, but a wrap-up sentence with references was added to discussion: “Survival in pediatric SAA improved dramatically for MUD HCT, due to improvements in donor typing, less toxic conditioning regimens with low dose TBI or TBI free and use of leukodepleted blood products.”
- What was the degree of HLA matches in the recipients of MUD?Ie how many were 10/10,9/10 etc.
The data are now presented in table 1.
- How come PBSC was used as graft sorce in SAA patients, who do not benefit from chronic GVHD in contrast to patients with malignant disorders.BM should be preferred in patients with SAA and other non-malignant diseases. Refer to studies showing a higher risk of chronic GVHD using PBSC vs BM and discuss pros and cons in pediatric HCT for SAA.
We totally agree with the Reviewer that BM should be the main source in SAA- HCT, but in our center majority of transplant is performed using PBSC. The trend towards increased number of PBSC transplantations is due to insufficient bone marrow harvests, that were complicating transplantations from unrelated donors in the past and need for graft processing leading to cell loss.
We added the following chapter to the discussion: “PBSC is not considered as the best stem cell source for the first HCT in SAA because of lower survival and higher risk of aGVHD and cGVHD [31,48,49]. However, more recent studies report better outcomes and similar results between between MUDs with PBSC or BM[12]. PBSC has been reported in cases of a second donation by the same donor where previously BM was used to improve engraftment and achieve faster hematopoietic recovery[50]. Horan et al. reported the incidence of graft failure of 43% in 166 SAA patients who underwent second HCT from MSD using BM or PBSC in 84% and 16% of patients, respectively[51]. Similarly in a pediatric study by Cesaro concerning second allogenic transplantation, use of PBSC wasn’t associated with inferior OS[52].”
- Please give more information in the table on characteristics like CMV status of recipients and donors, nucleated cell dose, CD 34+ celldose, sex match of donor and recipients etc. Such data is of interest for the reader to evaluate the results.
The data are now presented in table 2. The donor gender showed impact on the OS in univariate analysis, and the figure was added, but this effect was not present in COX model.
a paragraph on this observation was added to discussion: “Among different factors affecting OS, an impact of female donors (independent from recipient sex) was associated with inferior survival. Among non-HLA donor characteristics sex mismatching (male recipient-female donor) is a proven risk factor for inferior survival associated with cGVHD incidence, but due to low number of events in our study, this result must be approached with caution.”
- Regarding ATG, Thymoglobuline treatment, the patients were given different dosing and timing up to day -1 and day -3.Please discuss the importance of dosing and timing of Thymoglobuline with adequate references. This may affect risks of acute GVHD and infections respectively.
The addition of ATG is known to significantly reduce the risk of cGvHD and graft failure. Analyzed patients since 2011 were treated from day -6 to -3 with ATG Grafalon (Neovii) at daily dose of 15 mg/kg BW daily or Thymoglobulin (Sanofi) at 2.5 mg/kg BW daily. On this occasion, we have noted and corrected a clerical error with Grafalon daily dose (should be 15 instead of 20 mg/kg). We have referred papers on the importance of ATG TDM, although we have not performed the drug level monitoring.
A paragraph was added: “ATG has a narrow therapeutic window and therapeutic drug monitoring of ATG levels helps to optimize dosing to ensure timely T-cell immune reconstitution[30,31]. Exposure to ATG affects survival after HCT in adults, stressing the importance of optimum ATG dosing. According to these studies overexposure of ATG delays T-cell reconstitution and is associated with increased relapse rates and viral reactivations, whereas underexposure with incidence of GvHD and higher mortality[32]. Individualized dosing of ATG, based on lymphocyte counts rather than bodyweight in adults has been recommended by Admiraal at al., but in children higher exposure can be observed in patients with a higher bodyweight and/or a lower lymphocyte count pre-Thymoglobulin infusion[30,32]. “
- Minor, change HSCT to HCT, because this is not a true stemcell transplant.
The term “hematopoietic stem cell transplantation (HSCT)” was recommended by EBMT society and widely used in PubMed over 70000 references which is almost 15 times (5000 hits) more than hematopoietic cell transplantation (HCT), but to acknowledge contemporary nomenclature, we changed the term according to the reviewer’s remark.
- Supplementary data may be deleted, because they were ns. Furthermore they were not possible to download.
The supplementary data were compressed (by MDPI publisher) as ZIP archive, but can be viewed, at least from our side.
